# Rapid, Efficient, and Cost-Effective Gene Editing of *Enterococcus faecium* with CRISPR-Cas12a

Michelle J. Chua,[a] ⓘ James Collins[a]

[a]Department of Microbiology and Immunology, School of Medicine, University of Louisville, Louisville, Kentucky, USA

**ABSTRACT** Considered a serious threat by the Centers for Disease Control and Prevention, multidrug-resistant *Enterococcus faecium* is an increasing cause of hospital-acquired infection. Here, we provide details on a single-plasmid CRISPR-Cas12a system for generating clean deletions and insertions. Single manipulations were carried out in under 2 weeks, with successful deletions/insertions present in >80% of the clones tested. Using this method, we generated three individual clean deletion mutations in the *acpH*, *treA*, and *lacL* genes and inserted codon-optimized *unaG*, enabling green fluorescent protein (GFP)-like fluorescence under the control of the trehalase operon. The use of *in vivo* recombination for plasmid construction kept costs to a minimum.

**IMPORTANCE** *Enterococcus faecium* is increasingly associated with hard-to-treat antibiotic-resistant infections. The ability to generate clean genomic alterations is the first step in generating a complete mechanistic understanding of how *E. faecium* acquires pathogenic traits and causes disease. Here, we show that CRISPR-Cas12a can be used to quickly (under 2 weeks) and cheaply delete or insert genes into the *E. faecium* genome. This substantial improvement over current methods should speed up research on this important opportunistic pathogen.

**KEYWORDS** CRISPR-Cas12a, CRISPR, *Enterococcus faecium*, genome editing, counterselection, VRE

**E**nterococcus faecium is a Gram-positive commensal bacterium increasingly associated with antibiotic-resistant, hospital-associated infections. The ability of *E. faecium* to readily acquire multidrug resistance has led to its inclusion in the WHO global priority list of multidrug-resistant pathogens and its consideration as a serious threat by the CDC.

Efficient genome editing tools are essential for functional studies in emerging pathogens. Traditional methods to generate targeted mutations in *E. faecium* include an allelic exchange between the chromosome and a temperature-sensitive vector and the use of a *p*-chlorophenylalanine sensitivity counterselection system (1–3). Although capable of generating clean genetic manipulations, the time from vector construction to genetic manipulation can take 5 to 6 weeks and requires screening of many potential mutants.

In recent years, there have been considerable developments in genome editing tools that use CRISPR-Cas systems (i.e., systems involving clustered regularly interspaced short palindromic repeat-associated proteins) (4, 5). Initially identified as an adaptive immune system in bacteria and archaea, the critical functionality of CRISPR-Cas systems is their ability to generate a double-strand break (DSB) in DNA at a specific locus (6, 7). The introduction of a DSB into the genome is often fatal; genome editing can therefore be achieved through either the nonhomologous end joining (NHEJ) or homologous recombination (HR) mechanisms (8). Due to the limited ability for NHEJ in

Address correspondence to James Collins, james.collins.1@louisville.edu.

The authors declare no conflict of interest.

**FIG 1** Construction of the CRISPR-Cas12a knockout (KO) plasmid. The universal chassis plasmid, pJC005, was built to facilitate the construction of gene knockout or knock-in plasmids. pJC005 contains Cas12a from *Acidaminococcus* under the control of a tetracycline-inducible promoter, replication machinery for replication in both *E. coli* and *E. faecium*, and either erythromycin or gentamicin resistance cassettes. To generate pJC005.X*lacL*, enabling clean deletion of the 1.9-kb β-galactosidase gene (*lacL*), the small RNA promoter (sRNAP) fused to a LacL-specific spacer sequence is amplified from pUC19sRNAP and joined to the upstream (US) and downstream (DS) arms by SOE PCR. The product from the SOE PCR is then inserted into linearized pJC005 by *in vivo* cloning.

bacteria, CRISPR-Cas systems are typically used to select for Campbell-like integrations of heterologous DNA.

Recently, V. de Maat et al. (9) developed a two-plasmid CRISPR-Cas9-mediated genome editing system that reduced the time frame for genome editing in *E. faecium* to ~3 weeks and significantly reduced the amount of screening required. Although a significant improvement on legacy techniques, CRISPR-Cas9 has several limitations. CRISPR-Cas9 systems often require multiple (or large single) vectors, and the inherent toxicity of Cas9 can limit transformation efficiency (10). As an alternative to CRISPR-Cas9, CRISPR-Cas12a has been utilized in multiple bacterial species, including *Escherichia coli*, *Bacillus subtilis*, *Yersinia pestis*, *Corynebacterium glutamicum*, *Mycobacterium smegmatis*, and *Clostridioides difficile* (10–13). The Cas12a nuclease has several advantageous features. (i) Whereas Cas9 requires two RNA molecules, tracrRNA (*trans*-activating CRISPR RNA) and a crRNA (CRISPR RNA), Cas12a requires only a single RNA molecule, the crRNA. This results in a smaller vector size and allows for multiplexed genetic manipulation (12–15). (ii) The Cas12a protein recognizes a T-rich protospacer adjacent motif (PAM) sequence (5′-TTTV-3′), a common motif in low-GC microorganisms. (iii) Finally, it has been reported that Cas12a has lower toxicity and reduced off-target effects compared to Cas9 for genome editing (10, 15, 16).

To overcome the toxic effects of CRISPR-Cas9, limit the need for antibiotic selection in a two-plasmid system, and expedite genetic manipulation of *E. faecium*, we generated a single-plasmid CRISPR-Cas12a system capable of clean deletion and insertion of DNA in *E. faecium*: pJC005 (Fig. 1).

## RESULTS

**Genome editing in *E. faecium* with inducible CRISPR-Cas12a.** To determine if we could generate clean deletions efficiently, we independently targeted three genes in *E. faecium* NCTC7171: *lacL*, *acpH*, and *treA*. Details of each plasmid construct

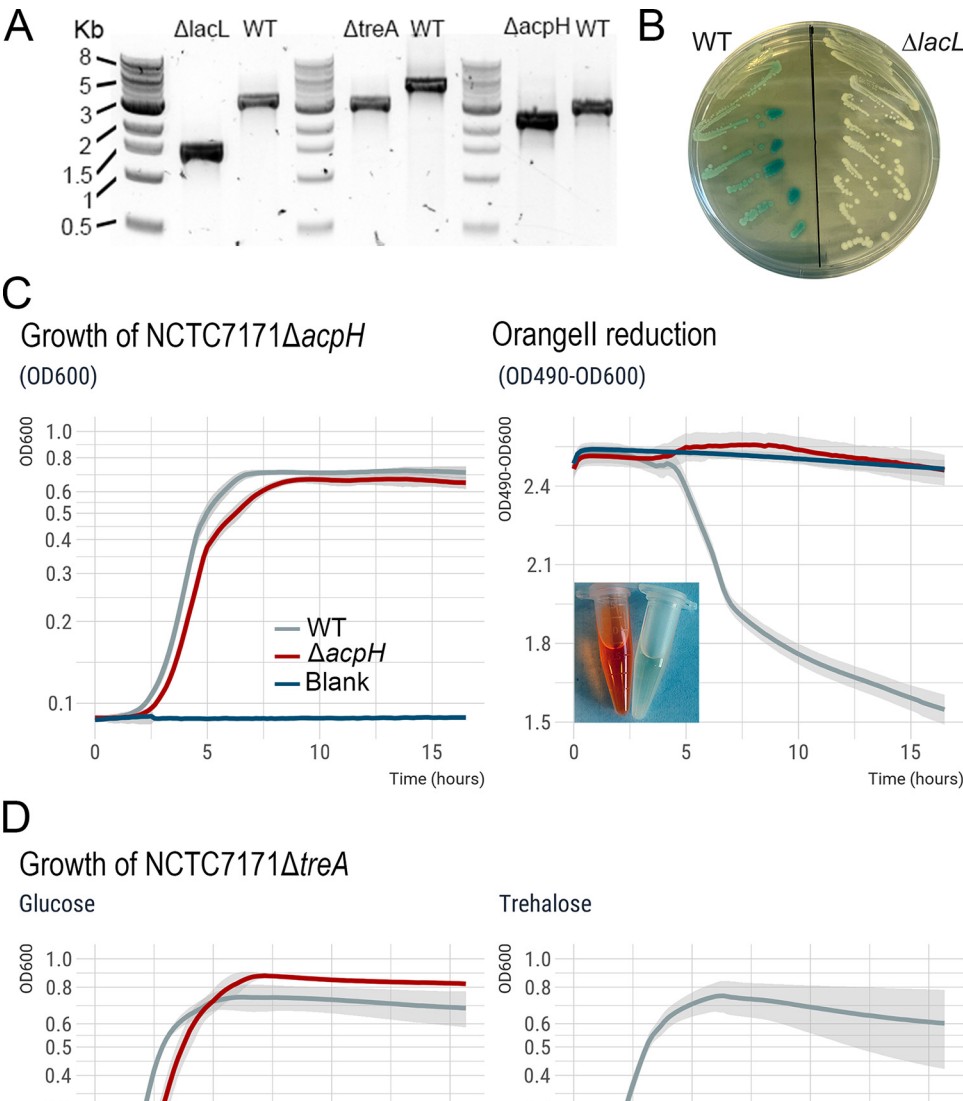

**FIG 2** Clean deletion of *E. faecium* genes. (A) Confirmation of *lacL*, *treA*, and *acpH* deletions. (B) Growth of wild-type NCTC7171 and the Δ*lacL* mutant on BHI with 20 µg mL$^{-1}$ X-Gal. (C) Reduction of Orange II by the azoreductase AcpH. (D) Growth in M1 medium supplemented with 0.5% (wt/vol) glucose or trehalose. Growth curves show the mean growth of three biological replicates, with shading representing the standard deviation.

(pJC005.em.X*lacL*, pJC005.em.X*acpH*, and pJC005.em.X*treA*) are given in the supplemental material. Following transformation of the knockout plasmids and induction of Cas12a, successful clean deletions were confirmed by colony PCR using primer pairs oJC079-oJC080, oMC060-oMC061, and oJC072-oJC073, respectively (Fig. 2A). We obtained clean deletions for each mutant within 2 weeks.

Wild-type (WT) and Δ*lacL* mutant strains were grown on brain heart infusion (BHI) supplemented with the chromogenic substrate X-Gal (5-bromo-4-chloro-3-indolyl-β-D-galactopyranoside) to confirm that the genomic alteration affected β-galactosidase activity. While WT colonies were blue upon growth, the NCTC7171 Δ*lacL* colonies

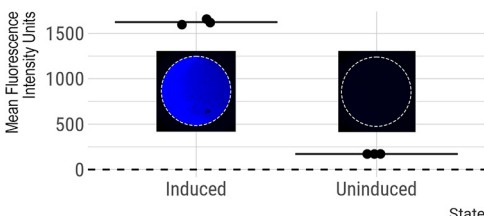

**FIG 3** Insertion of codon-optimized *unaG* under the control of the trehalose operon. UnaG is significantly upregulated following the addition of trehalose (Welch two-sample *t* test, *P* < 0.001) (*R*). (Insets) Induced (left) and uninduced (right) cells in a 96-well plate. The dashed horizontal line indicates the background fluorescent intensity of blank wells.

remained white, indicating that they could no longer convert X-Gal due to the lack of an active *β*-galactosidase (Fig. 2B). *Enterococcus faecium* possesses a 206-amino-acid azoreductase (encoded by a gene annotated as *acpH* in NCTC7171) capable of reducing certain azo dyes (17). We propagated *E. faecium* NCTC7171 in BHI containing 25 $\mu$M either methyl red, Congo red, or Orange II to confirm an azo dye reducing capability. While methyl red and Congo red showed no sign of breakdown either anaerobically or aerobically, NCTC7171 reduced Orange II under anaerobic conditions (Fig. 2C). To ensure this color change was not due to a pH change, we adjusted the pH of BHI plus 25 $\mu$M Orange II from 7 to 2.5 with glacial acetic acid. No color change was observed. Thus, *E. faecium* NCTC7171 is capable of reducing Orange II under anaerobic conditions. WT NCTC7171 and NCTC7171 Δ*acpH* were grown anaerobically in liquid BHI supplemented with 25 $\mu$M Orange II. While the WT strain could break down Orange II, the mutant strain could not. No differences in growth characteristics were observed (Fig. 2C). Finally, the WT and Δ*treA* strains were grown in M1 medium supplemented with 0.5% (wt/vol) trehalose or glucose. While both strains grew similarly on the glucose-supplemented medium, the Δ*treA* mutant was incapable of growth when provided with trehalose as the sole carbon source (Fig. 2D).

To test the ability of our system to make clean knock-ins, we inserted a codon-optimized bilin-binding fluorescent protein gene, *unaG*, in place of *treA*, under the control of the trehalase operon by using plasmid pJC005.em.X*treA*::*unaG*. (Detailed plasmid construction is provided in the supplemental material.) UnaG fluoresces only after the addition of cell-permeable bilirubin and, unlike many fluorophores, does not require oxygen, making it helpful in studying members of the gut microbiota (18). UnaG is maximally excited at 495 nm when bound to bilirubin, with a maximal emission at 525 nm. When grown in the presence of 0.5% (wt/vol) trehalose and 25 $\mu$M bilirubin, *E. faecium* NCTC7171 Δ*treA*::*unaG* had increased fluorescence (a 9.5-fold increase over noninduced controls). However, we observed no fluorescence in the absence of trehalose due to a lack of UnaG production (Fig. 3). Because oxygen is not required for fluorescence, this low-GC codon-optimized construct will likely be helpful for the future study of *E. faecium* in anaerobic environments like the gut.

## DISCUSSION

Infection with antibiotic-resistant enterococci has become a severe health problem, leading to thousands of deaths each year (19, 20). A fuller mechanistic understanding of *E. faecium* is needed; however, the tools to easily, quickly, and cheaply generate genomic alterations are lacking. To address this, we developed a single-plasmid CRISPR-Cas12a system to select for mutants following homologous recombination. The system can generate clean deletions and gene knock-ins with an efficiency of >80%. Following genetic manipulation, the plasmid is cleared within three passages in the presence of anhydrous tetracycline (ahTC). The Cas12a system likely targets the plasmid following the double-crossover event, reducing the time needed to clear the

**TABLE 1** Strains used in this study

| Strain | Description | Source or reference |
|---|---|---|
| *E. coli* | | |
| NEB 10-beta | Derivative of DH10B (*ara leu*)7697 *araD139 fhuA* Δ*lacX74 galK16 galE15 e14-*ϕ*80dlacZ*ΔM15 *recA1 relA1 endA1 nupG rpsL* (Str^r^) *rph spoT1* Δ(*mrr-hsdRMS-mcrBC*) | NEB |
| *E. faecium* | | |
| NCTC7171 | *E. faecium* type strain | ATCC |
| NCTC7171 Δ*treA* | NCTC7171 with clean deletion of *treA* | This study |
| NCTC7171 Δ*lacL* | NCTC7171 with clean deletion of *lacL* | This study |
| NCTC7171 Δ*acpH* | NCTC7171 with clean deletion of *acpH* | This study |
| NCTC7171 Δ*treA::unaG* | NCTC7171 with clean deletion of *treA* and insertion of *unaG* | This study |

plasmid in the absence of antibiotic selection. In total, the time from the arrival of primers to a plasmid-cured clean deletion can take less than 2 weeks, a substantial improvement over other techniques. Significant cost savings were achieved by utilizing *in vivo* cloning to replace comparable techniques, such as Gibson cloning or NEBuilder HiFi assembly.

Possible limitations of this system include an inability to assemble fragments toxic to *E. coli* due to the high-copy-number origin of replication. The toxicity may be overcome by replacing the pUC origin with a lower-copy-number *ori* or assembling plasmid constructs *ex vivo*. Furthermore, as with other techniques that rely on homologous recombination and deletion, this technique does not enable the deletion of essential genes. However, the system does lend itself for conversion to a CRISPR interference (CRISPRi) vector by mutating the Cas12a so that it no longer generates DNA strand breaks but instead interferes with transcription (21). While not shown here, pJC005 constructs are likely capable of both multiplexed genetic manipulations, as demonstrated for similar systems in other species (12, 15, 22), and use in other Gram-positive organisms as the backbone is known to replicate in multiple species, including *Lactococcus*, *Streptococcus*, and *Lactobacillus* (23, 24).

## MATERIALS AND METHODS

**Strains, media, and growth conditions.** The bacterial strains used in this study are listed in Table 1. *E. faecium* was routinely grown at 37°C in brain heart infusion (BHI) medium supplemented as required with erythromycin (50 $\mu$g mL$^{-1}$) and anhydrous tetracycline (ahTC) (250 ng mL$^{-1}$) as needed. For growth on a single carbon source, M1 medium (10 g of tryptone and 0.5 g of yeast extract in 1 L of phosphate-buffered saline, adjusted to pH 7.4 with HCl) was used (25). As required, *Escherichia coli* strains were grown in lysogeny broth (LB) medium at 37°C supplemented with erythromycin (200 $\mu$g mL$^{-1}$).

**Plasmids and primers.** The plasmids used in this study are listed in Table 2. The primers used in this study are listed in Table 3. Q5 high-fidelity DNA polymerase (NEB) and *Taq* 2× master mix (NEB) were used for cloning and colony PCR, respectively. Plasmids were purified from *E. coli* using the Zyppy plasmid kit (Zymo). Transformation of plasmids into *E. faecium* NCTC7171 was performed as previously described, except cells were grown in BHI medium supplemented with 1% glycine and 300 mM sucrose (26).

**SOE PCR.** To increase the efficiency of *in vivo* cloning, the small RNA promoter containing the CRISPR spacer and the upstream and downstream homologous arms were first joined by splicing by overlap extension (SOE) PCR (27). Individual PCR products were cleaned by column purification. Fifty nanograms of the largest fragment and equimolar amounts of the smaller pieces were combined into a 20-$\mu$L final volume of Q5 PCR without primers. The NEB melting temperature ($T_m$) calculator was used to determine annealing temperatures for the following cycling conditions: 98°C for 30 s, then 10 cycles of 98°C for 10 s, a higher join $T_m$ for 10 s, and a lower join $T_m$ for 20 s, followed by 72°C for 1 min. Following the first round of SOE PCR, 1 $\mu$L of each universal SOE primer (oMC087-oMC088) was added, and the reaction mixture was returned to the thermocycler for 15 cycles of 98°C for 10 s, 65°C for 20 s, and 72°C for 1 min. One microliter of the SOE PCR product was used directly without cleanup for *in vivo* cloning if a single band was observed following gel electrophoresis. If multiple bands were present, the band of the correct size was gel extracted before use.

**In vivo cloning.** To knock out specific genes, pJC005 requires three inserts: a small RNA promoter driving the CRISPR protospacer RNA and up- and downstream arms homologous to the flanking regions of the target gene. To knock in a gene, the gene of interest is also required. *In vivo* cloning, as outlined by F. Huang et al. (28), was used to construct all deletion/insertion vectors. Briefly, pJC005.em was linearized by either PCR (primer pair oJC218-oJC219) followed by DpnI digest of template DNA or direct restriction digest using BtgZI. Linearized backbone (50 ng) and the SOE PCR product obtained as

**TABLE 2** Plasmids used in this study

| Plasmid | Feature(s) | Source or reference |
|---|---|---|
| pUC19 | pUC, *bla* | 33 |
| pUCsRNAP | pUC, *bla*; small RNA promoter and 23-bp spacer sequence flanked by two 19-bp repeats | This study |
| pRPF185 | Tetracycline-inducible expression plasmid | 34 |
| pDL-1 | pMTL82151 backbone with Cas12a under control of Ptet promoter from pRPF185 | Gift from Duolong Zhu, Baylor College of Medicine |
| pVPL3004 | Em$^r$ derivative of pNZ9530 | 24 |
| pJC005.em | Em$^r$ CRISPR-Cas12a genome editing plasmid | This study |
| pJC005.gent | Gm$^r$ CRISPR-Cas12a genome editing plasmid | This study |
| pJC005.em.X*acpH* | Em$^r$ plasmid targeting clean deletion of *acpH* | This study |
| pJC005.em.X*lacL* | Em$^r$ plasmid targeting clean deletion of *lacL* | This study |
| pJC005.em.X*treA* | Em$^r$ plasmid targeting clean deletion of *treA* | This study |
| pJC005.em.X*treA::unaG* | Em$^r$ plasmid targeting clean knock-in of *unaG* in place of *treA* | This study |

outlined above were transformed into chemically competent *E. coli* DB10 cells. Colonies were checked 24 h later for the correct insert by colony PCR. Details of plasmid construction are provided in the supplemental material.

**Construction of a CRISPR-Cas12a gene editing system for *Enterococcus faecium*.** W. Hong et al. (12) previously demonstrated the utility of a CRISPR-Cas12a system in the low-GC Gram-positive organism *C. difficile* (12). As *C. difficile* and *E. faecium* have similar codon usage, we utilized Cas12a placed under the control of the tetracycline-inducible promoter (Ptet) from pRPF185 on the pMTL82151 backbone (pDL1, a gift from Duolong Zhu, Baylor College of Medicine). The *slpA* terminator from pRPF185 was PCR amplified with primer pair oJC186-oJC187, containing a pair of BtgZI sites on the reverse primer, and cloned into the BamHI and XhoI sites. To enable replication in *E. faecium* and *E. coli* (as a host for *in vivo* cloning), we PCR linearized plasmid pVPL3004 (shown to replicate in *E. faecium* [9]) with primer pair oJC023-oJC024 and inserted a PCR-amplified 863-bp fragment (primer pair oJC025-oJC026) containing the pUC19 origin of replication. Site-directed mutagenesis with primer pair oJC195-oJC196 was used to remove the BtgZI site in *repE* of pVPL3004. Finally, a 5.5-kb fragment of the new shuttle vector backbone was linearized by PCR (oJC047-oJC048) and combined with a 5.5-kb PCR (oJC045-oJC046) segment containing Ptet-Cas12a-*thiT*-2×BtgZI. The newly constructed vector, pJC005.em (see Fig. S1 in the supplemental material), can replicate in *E. coli* and *E. faecium*. Usability was maximized by building a second plasmid, pJC005.gent, by replacing the erythromycin resistance cassette with the gentamicin resistance cassette from p34s-Gm (29). To enable quick changes of the CRISPR target sequence, we cloned a synthetic 483-bp DNA fragment consisting of a small RNA promoter and a 23-bp spacer sequence flanked by two 19-bp repeats into pUC19, resulting in pUC19.pcRNA (see the supplemental material for details). CRISPR targets were designed by identifying 23-nucleotide (nt) sequences directly downstream of a 5′-TTTV-3′ PAM site (30). To generate a clean deletion or insertion vector, the small RNA promoter fused to a target spacer sequence and the up- and downstream homologous arms (as well as an

**TABLE 3** Primers used in this study

| Name | Purpose | Sequence 5′→3′ |
|---|---|---|
| oJC023 | pVPL3004 linearization | AAGTATATATGAGTAAAGATGTGATCCGTAGCGGT |
| oJC024 | | TTGAGTGAGCGTTGGAACCATTCTTAACAGCA |
| oJC025 | pUC19 origin of replication | AAGAATGGTTCCAACGCTCACTCAAAGGCGGTAATAC |
| oJC026 | | ATCACATCTTTACTCATATATACTTTAGATTGATTTAAAAC |
| oJC045 | Ptet-Cas12a-*thiT*-2×BtgZI | ATCCGTAGCGGTATGCGCTCCATCAAGAAGAG |
| oJC046 | | ATTCCTGGTTGCTCCAGGGTGCTATCTTCGTC |
| oJC047 | New shuttle vector backbone linearization | TAGCACCCTGGAGCAACCAGGAATGAATTACTATCCC |
| oJC048 | | GATGGAGCGCATACCGCTACGGATCACATCTT |
| oJC072 | Confirm Δ*treA* | TTGAAGAGAACCCACGCTCC |
| oJC073 | | CCCCGTGTTCATGGTCATGA |
| oJC079 | Confirm Δa*cpH* | ATTGGAGGTAAGTGTCGGCA |
| oJC080 | | GACCCGCTGTTCAACCAATT |
| oJC186 | pRPF185 *slpA* terminator | CAACAACTCTCCTGGCGCAC |
| oJC187 | | ATACTCGAGTCATCGCTGACTCATGCGATGACGAATTCCAGCACACTGGCA |
| oJC218 | pJC005.em linearization | CTCGAGGCCTGCAGACATGC |
| oJC219 | | GACGAATTCCAGCACACTGGCATC |
| oMC060 | Confirm Δ*lacL* | TCGTTATATGCTCGGGCTTT |
| oMC061 | | AAAGCAAATGGCGTATCCTG |
| oMC087 | SOE PCR universal primers | GAGAAATCCCTAAATAAAAAGATGCCAGTGTGCTG |
| oMC088 | | GCCAGTGCCAAGCTTGCATGTCTGCAGGCCTCGAG |

insertion sequence if used) were joined by splice by overlap extension (SOE) PCR and combined with linearized pJC005 by *in vivo* cloning within *E. coli* DB10 cells (Fig. 1).

**Generation of clean deletion/insertion mutants in *E. faecium*.** Following transformation of the CRISPR knockout or knock-in plasmid into *E. faecium*, a single colony was picked into 5 mL of prewarmed BHI medium supplemented with 50 $\mu$g mL$^{-1}$ erythromycin and propagated with shaking at 37°C for at least 8 h followed by a 1:1,000 dilution into fresh BHI medium and propagation overnight. The following day, 1 $\mu$L of culture was struck onto BHI plates supplemented with 50 $\mu$g mL$^{-1}$ erythromycin and 250 ng mL$^{-1}$ ahTC and incubated at 37°C. Sixteen colonies were randomly picked for colony PCR to screen for mutants. Over 80% of the screened colonies were confirmed to have the correct insertion/deletion in all cases. The Cas12a plasmid was cured by transferring a colony with the desired mutation to 5 mL BHI medium plus 250 ng mL$^{-1}$ ahTC without antibiotics and incubated overnight. Concurrently, the cultures were subcultured 1:1,000 into fresh BHI medium plus 250 ng mL$^{-1}$ ahTC, and 1 $\mu$L of culture was struck onto a BHI plate. Following growth, colonies were patch plated with and without erythromycin to determine plasmid loss. The majority of screened colonies showed plasmid loss after one passage, and within three passages, 100% of the screened colonies had cured the plasmid.

**Plate reader growth and Orange II breakdown analysis.** Three independent biological replicates were used to confirm the phenotypes of all mutant strains. NCTC7171 and NCTC7171 Δ*treA* were incubated overnight at 37°C in BHI medium. The following morning, cultures were diluted back to an optical density at 600 nm (OD$_{600}$) of 0.05 in M1 medium supplemented as required with 0.5% (wt/vol) trehalose or glucose. Two hundred microliters was added to a 96-well plate in triplicate technical replicates, and OD$_{600}$ readings were taken every 10 min for 16 h in an anaerobic microplate reader.

To determine the azoreductase activity of the Δ*acpH* mutant, NCTC7171 and NCTC7171 Δ*acpH* were incubated overnight at 37°C in BHI medium. The following morning, cultures were diluted to an OD$_{600}$ of 0.05 in fresh BHI supplemented with 25 $\mu$M Orange II. Two hundred microliters of cultures was added to a 96-well plate in triplicate technical replicates. Strain growth and breakdown of Orange II were measured by taking OD$_{600}$ and OD$_{490}$ readings, respectively, every 10 min for 16 h in an anaerobic microplate reader.

**Analysis of UnaG fluorescence in NCTC7171.** Expression of the codon-optimized *unaG* gene was confirmed using the Sapphire biomolecular imager. NCTC7171 and NCTC7171 Δ*treA*::*unaG* were incubated at 37°C in BHI medium overnight. Biological triplicates were diluted 1:24 in M1 medium supplemented with 25 $\mu$M bilirubin and 0.25% glucose and split into induced and uninduced tubes. The induced tubes received 0.5% (wt/vol) trehalose, while 0.5% (wt/vol) glucose was added to the uninduced tubes. Cultures were incubated for 8 h at 37°C with shaking before cells were harvested by centrifugation at 3,000 $\times$ *g* for 5 min. Cell pellets were resuspended in 1 mL M1 medium, 200 $\mu$L was added to a 96-well plate in technical triplicates, and the mixture was centrifuged at 3,000 $\times$ *g* for 5 min. The Sapphire biomolecular imager was used to image the plate with the following settings: 100-$\mu$m pixel size, a focal plane of 3 mm above the glass scanning surface, excitation at 488 nm, and emission filter 518BP22. Fluorescence intensity was calculated with ImageJ (31). The Welch two-sample *t* test was used to test for a significant difference between groups by using the R programming language (32).

## SUPPLEMENTAL MATERIAL

Supplemental material is available online only.
**SUPPLEMENTAL FILE 1**, PDF file, 0.5 MB.

## ACKNOWLEDGMENTS

This work was supported in part by a grant from the Jewish Heritage Fund for Excellence Research Recruitment Grant Program at the University of Louisville, School of Medicine, and Centers of Biomedical Research Excellence (COBRE) Grant GM125504.

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
