## [Reviewer comments · Microbiology Spectrum]

Microbiology Spectrum

Rapid, efficient, and cost-effective gene editing of *Enterococcus faecium* with CRISPR-Cas12a

Michelle Chua and James Collins

Corresponding Author(s): James Collins, University of Louisville

Review Timeline:

Submission Date:	November 29, 2021
Editorial Decision:	December 20, 2021
Revision Received:	January 4, 2022
Accepted:	January 7, 2022

Editor: Justin Kaspar

Reviewer(s): Disclosure of reviewer identity is with reference to reviewer comments included in decision letter(s). The following individuals involved in review of your submission have agreed to reveal their identity: Robert Shields (Reviewer #2)

Transaction Report:

DOI: <https://doi.org/10.1128/spectrum.02427-21>

December 20, 2021

Dr. James Collins
University of Louisville
Department of Microbiology & Immunology
Louisville, KY

Re: Spectrum02427-21 (Rapid, efficient, and cost-effective gene editing of *Enterococcus faecium* with CRISPR-Cas12)

Dear Dr. James Collins:

Two expert reviewers have provided opinions on your manuscript. Both are enthusiastic about the quality and significance of the work, and consider it to be appropriate for ASM Spectrum. The reviewers have a few small suggestions for improvement of the presentation, all of which should be easily implemented. I think it will be especially important to add further detail where requested by the reviewers and additional labeling / modifications to the figures as suggested. Please find all of the reviewer comments at the bottom of this email.

Link Not Available

Sincerely,

Justin Kaspar

Journals Department
Reviewer comments:

Reviewer #1 (Comments for the Author):

This is an exciting manuscript by Chua and Collins detailing an advance in genetic manipulation tools for *Enterococcus faecium*. The authors developed a CRISPR-Cas12 single plasmid system and demonstrated that it can be used to rapidly knock out genes in *E. faecium*. As proof of concept, they show 3 examples of gene deletions and phenotypes associated with each deletion. Overall, this new system is important for not only *Enterococcus faecium*, but also other Gram-positive organisms for which abundant gene manipulation tools do not yet exist.

I have minor comments regarding the content of the manuscript:

Lines 28-31 (and first sentence of the discussion): citations are needed.

Lines 154-156: the pH data should be added.

Lines 150-164: this is minor, but it's slightly confusing for the text to go from Figure 2C to 2B to 2D, please consider swapping the order the panels are introduced to keep it consistent with how the figure is laid out.

Line 163: change "thrived" to "grew similarly"

Methods and Figure 3: please name the program used for statistical analysis

Line 171: change "is seen fluorescing" to "had increased fluorescence" (or reword accordingly)

Line 188: could remove *Enterococcus* from this list (redundant because the this paper already describes the tool in *Enterococcus*)

Discussion: could the authors speculate on factors that could complicate using this CRISPR-based system over existing genetic manipulation techniques? For example, how much time do the authors think this system could save over existing tools for mutants that have growth defects? Also, pUC19 is high copy so some constructs may not be tolerated in *E. coli*.

Figure S1: this figure is useful but could benefit from more specific labeling. What are the gray arrows? Please label the region where the sRNA promoter-target sequence are combined with in vivo cloning.

Table 3: it would be helpful to have labels showing which gene knockouts were made with which primers

Table 3/Supp Table: it's unclear what primers are in the main table vs supplementary table as there is some overlap (some primers found in both tables)

Table 4: I am not sure this is necessary, as the timeline will probably shift if gene deletion mutants grow slowly or the initial plasmid construct isn't easily obtained by in vivo cloning. In this reviewer's opinion, the exact time saved over existing techniques is secondary to the overall importance of the development of a new genetic manipulation technique for an understudied organism.

Reviewer #2 (Comments for the Author):

The manuscript by Chua and Collins describes an improved methodology for genetic manipulation in *E. faecium* using CRISPR-Cas12a. Although a CRISPR-Cas9-based genome-editing system has been developed in *E. faecium*, the authors argue that a CRISPR-Cas12a system has notable benefits. These include a simpler design with one plasmid (versus two in the Cas9-based system) and expedited cloning (2 weeks instead of 3 weeks). It is also possible that Cas12a is less toxic to *E. faecium* based on observations in other microbial species. The authors logically design a streamlined system for CRISPR-Cas assisted cloning in *E. faecium*, with spacer target modification relying on SOE-PCR and in vivo cloning. Evidence of the utility of the system is shown in several ways: (i) deletion of *lacL*, *treA*, and *acpH*, (ii) loss of beta-galactosidase activity in the *lacL*- strain, (iii) growth defect of the *treA*- strain with trehalose as the carbon source, (iv) loss of *acpH* activity, (v) and knock-in of *unaG* in place of *treA* with fluorescence readings to confirm. Overall, on the basis of the data shown, the system works and should be beneficial to other *E. faecium* researchers. As the authors mentioned, the plasmid should also work in closely related bacteria, including other enterococci and streptococci. The manuscript is already of good quality, and I have only a few minor comments:

1. CRISPR-Cas12 and CRISPR-Cas12a are both used in the manuscript - it might be cleaner to only use Cas12a.
2. It wasn't clear to me if after the spacer target modification step (SOE-PCR) if the in vivo cloning into linearized backbone is performed in *E. coli* or *E. faecium* - first mentioned line 108. This information is available in the supplemental material but should also appear in the main text.
3. Line 73. I think it would be worthwhile to define SOE-PCR and potentially add a reference. This might help readers who are new to cloning understand this important part of the methodology.
4. It is not very clear how many PCRs are needed to make the CRISPR-target region that is subsequently cloned into the plasmid. I believe it is PsRNA-spacer, US-arm and DS-arm that are combined together with SOE-PCR. I think Figure 1 could be modified to make this clearer. Also, text that is currently in the supplemental material could be moved to the methods section as the explanation is clearer there e.g. "to knock-out specific genes, pJC005 requires three inserts...". Clarity is really important for helping others with limited cloning experience to understand the system.

Staff Comments:

Preparing Revision Guidelines

Please return the manuscript within 60 days; if you cannot complete the modification within this time period, please contact me. If you do not wish to modify the manuscript and prefer to submit it to another journal, please notify me of your decision immediately so that the manuscript may be formally withdrawn from consideration by Microbiology Spectrum.

Dear reviewers,

Thank you for taking the time to review our work and make constructive comments that improved the final paper. We provide a summary of the response below:

Reviewer #1

Lines 28-31 (and first sentence of the discussion): citations are needed.

Multiple citations added.

Lines 154-156: the pH data should be added.

We have added the pH data

Lines 150-164: this is minor, but it's slightly confusing for the text to go from Figure 2C to 2B to 2D, please consider swapping the order the panels are introduced to keep it consistent with how the figure is laid out.

Text has been rearranged for constancy.

Line 163: change "thrived" to "grew similarly"

Done.

Methods and Figure 3: please name the program used for statistical analysis

ImageJ and R information and citations have been added.

Line 171: change "is seen fluorescing" to "had increased fluorescence" (or reword accordingly)

Done.

Line 188: could remove Enterococcus from this list (redundant because the this paper already describes the tool in Enterococcus)

Removed.

Discussion: could the authors speculate on factors that could complicate using this CRISPR-based system over existing genetic manipulation techniques? For example, how much time do the authors think this system could save over existing tools for mutants that have growth defects? Also, pUC19 is high copy so some constructs may not be tolerated in E. coli.

Potential pitfalls and ways around them were added to the discussion.

Figure S1: this figure is useful but could benefit from more specific labeling. What are the gray arrows? Please label the region where the sRNA promoter-target sequence are combined with in vivo cloning.

Fig S1 has been improved for clarity.

Table 3: it would be helpful to have labels showing which gene knockouts were made with which primers

Extra column added for primer labels

Table 3/Supp Table: it's unclear what primers are in the main table vs supplementary table as there is some overlap (some primers found in both tables)

Tables have been improved, and duplicates removed

Table 4: I am not sure this is necessary, as the timeline will probably shift if gene deletion mutants grow slowly or the initial plasmid construct isn't easily obtained by in vivo cloning. In this reviewer's opinion, the exact time saved over existing techniques is secondary to the overall importance of the development of a new genetic manipulation technique for an understudied organism.

We agree and have removed Table 4.

Reviewer #2 (Comments for the Author):

1. CRISPR-Cas12 and CRISPR-Cas12a are both used in the manuscript - it might be cleaner to only use Cas12a.

Cas12a is now used throughout.

2. It wasn't clear to me if after the spacer target modification step (SOE-PCR) if the in vivo cloning into linearized backbone is performed in *E. coli* or *E. faecium* - first mentioned line 108. This information is available in the supplemental material but should also appear in the main text.

This has now been clarified in the text

3. Line 73. I think it would be worthwhile to define SOE-PCR and potentially add a reference. This might help readers who are new to cloning understand this important part of the methodology.

Certainly worthwhile, definition and citation added.

4. It is not very clear how many PCRs are needed to make the CRISPR-target region that is subsequently cloned into the plasmid. I believe it is PsRNA-spacer, US-arm and DS-arm that are combined together with SOE-PCR. I think Figure 1 could be modified to make this clearer. Also, text that is currently in the supplemental material could be moved to the methods section as the explanation is clearer there e.g. "to knock-out specific genes, pJC005 requires three inserts...". Clarity is really important for helping others with limited cloning experience to understand the system.

Both the text and Figure have been modified to increase clarity.

January 7, 2022

Dr. James Collins
University of Louisville
Department of Microbiology & Immunology
505 S Hancock
Louisville, KY

Re: Spectrum02427-21R1 (Rapid, efficient, and cost-effective gene editing of *Enterococcus faecium* with CRISPR-Cas12a)

Dear Dr. James Collins:

Thank you for addressing all reviewer comments and editing the figures/text as suggested. Your manuscript has been accepted, and I am forwarding it to the ASM Journals Department for publication. You will be notified when your proofs are ready to be viewed.

Sincerely,

Justin Kaspar
Editor, Microbiology Spectrum
